# New Genomic Regions Identified for Resistance to Spot Blotch and Terminal Heat Stress in an Interspecific Population of *Triticum aestivum* and *T. spelta*

**DOI:** 10.3390/plants11212987

**Published:** 2022-11-05

**Authors:** Sudhir Navathe, Ajeet Kumar Pandey, Sandeep Sharma, Ramesh Chand, Vinod Kumar Mishra, Dinesh Kumar, Sarika Jaiswal, Mir Asif Iquebal, Velu Govindan, Arun Kumar Joshi, Pawan Kumar Singh

**Affiliations:** 1Institute of Agricultural Sciences, Banaras Hindu University, Varanasi 221005, India; 2Agharkar Research Institute, G.G. Agharkar Road, Pune 411004, India; 3Centre for Agricultural Bioinformatics, ICAR-Indian Agricultural Statistics Research Institute, Library Avenue, PUSA, New Delhi 110012, India; 4Department of Biotechnology, School of Interdisciplinary and Applied Sciences, Central University of Haryana, Mahendergarh 123031, India; 5International Maize and Wheat Improvement Center (CIMMYT), Veracruz 56237, Mexico; 6Borlaug Institute for South Asia, NASC Complex, DPS Marg, New Delhi 110012, India; 7International Maize and Wheat Improvement Center (CIMMYT), G-2, B-Block, NASC Complex, DPS Marg, New Delhi 110012, India

**Keywords:** *Bipolaris sorokiniana*, Indo–Gangetic plain, resistance, terminal heat

## Abstract

Wheat is one of the most widely grown and consumed food crops in the world. Spot blotch and terminal heat stress are the two significant constraints mainly in the Indo–Gangetic plains of South Asia. The study was undertaken using 185 recombinant lines (RILs) derived from the interspecific hybridization of ‘*Triticum aestivum* (HUW234) × *T. spelta* (H^+^26)’ to reveal genomic regions associated with tolerance to combined stress to spot blotch and terminal heat. Different physiological (NDVI, canopy temperature, leaf chlorophyll) and grain traits (TGW, grain size) were observed under stressed (spot blotch, terminal heat) and non-stressed environments. The mean maturity duration of RILs under combined stress was reduced by 12 days, whereas the normalized difference vegetation index (NDVI) was 46.03%. Similarly, the grain size was depleted under combined stress by 32.23% and thousand kernel weight (TKW) by 27.56% due to spot blotch and terminal heat stress, respectively. The genetic analysis using 6734 SNP markers identified 37 significant loci for the area under the disease progress curve (AUDPC) and NDVI. The genome-wide functional annotation of the SNP markers revealed gene functions such as plant chitinases, NB-ARC and NBS-LRR, and the peroxidase superfamily Cytochrome P450 have a positive role in the resistance through a hypersensitive response. Zinc finger domains, cysteine protease coding gene, F-box protein, ubiquitin, and associated proteins, play a substantial role in the combined stress of spot blotch and terminal heat in bread wheat, according to genomic domains ascribed to them. The study also highlights *T. speltoides* as a source of resistance to spot blotch and terminal heat tolerance.

## 1. Introduction

Wheat is one of the most widely grown and consumed food crops globally, having exceptionally high importance in the food system of South Asia. The eastern part of South Asia, which encompasses the eastern Gangetic Plains (EGP) of India, Nepal, and Bangladesh, is one of the most heavily populated parts of the world. In the EGP, where wheat is grown in about 10 m ha, the two major stresses to the wheat crop are spot blotch (SB) and terminal heat [1,2,3]. In EGP, SB caused by *Bipolaris sorokiniana* causes considerable yield loss between 15.5 and 19.6% annually [4]. However, when the disease is initiated at the flag leaf stage, losses of grains are estimated to be up to 24.2% [2]. (Singh et al., 2015). SB is normally a weak disease that takes advantage of heat stress (Rosyara et al., 2009), nutrient deficiency [5], and water stress [6] to induce significant grain damage. SB is favoured by cloudy and foggy days during the post-heading stage [7,8] and is expanding towards nontraditional cooler regions such as India’s North West Plain Zone (NWPZ) [2,4]. Further, this disease is predicted to become more severe due to climate change, nutritional and water deficiencies, and increased heat stress [9].

Terminal heat and spot blotch lead to premature leaf senescence, reduced grain filling, low kernel weight, and yield reductions [1,3]. NDVI has been used as an indirect criterion for stay-green and higher grain yield under drought or heat conditions and spot blotch resistance [10]. NDVI and yield associations have been well recorded [10,11,12]. Grain yield, controlled by several component traits, is important for overall production and food security. However, component traits are equally important for production and the market value and milling yield of bread wheat [13,14].

Due to their importance in affecting wheat production, breeding for heat stress tolerance and spot blotch resistance are the two critical objectives of wheat improvement programs targeting the EGP of South Asia. Genetic evaluation for heat tolerance and spot blotch in cultivated wheat has been attempted separately, and resistance sources have been identified [3,15,16,17,18]. However, limited information is available for the genomic regions providing tolerance to the combined stress of spot blotch and terminal heat stress. Hence, this study was initiated to identify the genomic regions associated with the combined stress of spot blotch and terminal heat, wherein a population derived from the cross of *T. aestivum* and *T. spelta* was utilized.

## 2. Results

### 2.1. Descriptive Statistics for Quantitative Traits Indicate Reductions in Yield Contributing Traits Due to the Combined Stress of SB and Terminal Heat

The distribution of 185 RILs and the parents for mean values for the nine phenotypic traits under different sowing dates and treatments—control, spot blotch, terminal heat stress, and combined stress spot blotch and terminal heat, is presented in Figure 1. The mean and range of phenotypic traits in the RILs across the environments and treatments are given in Appendix A. Under control conditions, the mean TGW was 34.17 ± 2.85, about 12% lower than the mean of 30.05 ± 3.33 g under SB infections. The decrease in grain area was observed from 11.57 ± 056 mm^2^ to 7.84 ± 0.53 mm^2^ (32.23%), which implicated in the reduction of thousand kernel weight from 34.17 ± 2.85 gm to 24.75 ± 2.46 gm (27.56%) (Table 1, Figure 1). The mean maturity duration (115.2 ± 1.19 days) under protected conditions decreased by >5% to 107.32 ± 4.06 days when infected to SB and further to 106.12 ± 1.22 and 103.76 ± 3.57 days under terminal heat stress and combined stress, respectively. The mean CT of the RILs under-protected was 23.11 ± 0.81 °C, while 25.73 ± 0.95 °C under spot blotch infection. A slight increase was noticed under terminal heat stress (30.23 ± 1.01 °C) and combined stress (31.75 ± 0.8 °C). The NDVI varied between 0.52–0.72, while after infection mean NDVI ranged between 0.39–0.59. It decreased to 0.42 ± 0.04 and 0.31 ± 0.03 units under terminal heat stress and combined SB + terminal heat stress (Appendix A).

The mean SPAD values were noted at 48.66 ± 2.4 units under protected conditions. However, this depleted to 43.43 ± 3.25 in response to SB infection. Under terminal heat stress, the SPAD mean was 48.63 ± 3.99, significantly decreasing to 26.1 under combined stress.

The AUDPC ranged between 299.31 and 689.35 with a mean of 504.69 7± 71.97. It was elevated to 731.14 ± 127.64 under combined stress. The decrease in grain area was 11.57 ± 056 mm^2^ to 7.84 ± 0.53 mm^2^ (32.23%) which was implicated in the reduction of TKW from 34.17 ± 2.85 gm to 24.75 ± 2.46 gm (27.56%) (Table 1).

The combined ANOVA for phenotypic traits indicated significant (*p* ≤ 0.0001) differences for a year, sowing condition, treatment, RILs, and their interactions (Table 2). A highly significant and positive correlation was found for NDVI with DH, TGW, DM, SPAD, grain area, and grain perimeter (Appendix A). A considerable grain area and grain perimeter correlation was obtained with DH, TGW, DM, SPAD, and NDVI, whereas AUDPC positively and significantly correlated with CT. Negative but highly significant associations were found between AUDPC with days to heading, TKW, DM, SPAD, NDVI, grain area, and grain perimeter.

### 2.2. Diversity and Population Structure Analysis by SNP and DArT Markers

The summary of minor allele frequency (MAF) and density of 5812 polymorphic SNP and DarT markers distributed on 21 chromosomes is given in Appendix A. The population STRUCTURE analysis over 187 lines revealed the presence of three populations. The proportion of each population in the three clusters was 0.612, 0.056, and 0.332, respectively, indicating that the three clusters contained 114, 11, and 62 genotypes. Average distances (expected heterozygosity) between individuals within each cluster (K1–K3) were 0.2243, 0.1352, and 0.2207, respectively. The net nucleotide distance among structures, i.e., the average probability that a pair of alleles was different between K1 vs. K2, K1 vs. K3, and K2 vs. K3, was 0.2591, 0.1190, and 0.2834, respectively. The mean value of alpha was observed at 0.0485. Further, for each cluster (K1–K3), the mean value of Fst was e 0.5572, 0.7292, and 0.5630, respectively (Appendix A).

The three-dimensional plot of the principal component analysis showing the genetic difference among RILs is shown in Figure 2a, while the heat map developed from 6734 SNP markers is in Figure 2b. The proportion and cumulative variances of the first three (3) PCAs were 16.80%, 6.10%, and 3.41%, respectively (Figure 2 and Appendix A).

Analysis of linkage disequilibrium: Out of 6734 markers used for the association mapping, 6369 markers were included in linkage disequilibrium analysis (LD). We filtered the markers with a minor allele threshold of 0.05, missing genotype 0.05, and removed individuals with a genotyping error of 0.1. (Appendix A). We arrived at 2611 markers from the whole genome, which were further obtained for LD analysis. A total of 129,276 locus pairs were detected, and 32,221 locus pairs (24.92%) were found to be in LD at *p* < 0.001, of which 23,281 locus pairs (72.25%) were found at r2 > 0.1 and *p* < 0.001 (Appendix A).

### 2.3. Marker Trait Analysis Identifies the Unique SNP and Candidate Genes for Terminal Heat Stress and Spot Blotch Resistance

Eighty-five (85) significant marker–trait associations were identified for the nine different phenotypic traits over four different environments (Figure 3; Appendix A). These marker–trait associations comprised thirty-seven (37) makers distributed majorly on nine chromosomes 1A, 1B, 2A, 3A, 5A, 5B, 6B, 7A, and 7B. The details of SNP makers and sequences are detailed in Appendix A.

The group of seven markers viz., 1125940|F|0 (1A), 1395486|F|0 (1B), 2256281|F|0, 980238|F|0 (3A), 1050819|F|0 (4D), 1029559|F|0, and 1020582|F|0 (5B) was commonly associated with the traits—grain area, days to heading, days to maturity, SPAD, and TKW. Seventeen identified markers are commonly associated with days to heading and days to maturity. For NDVI, nine unique marker–trait associations were identified, out of which two were on Chromosome 1A, three on 5A, and one each on 1B, 2A, 6B, and 7B. For AUDPC, five marker–trait associations were identified on chromosomes 2A, 5B, and 2D. The markers associated with SPAD were commonly associated with days to heading and days to maturity. Similarly, the markers associated with grain area, CT and TKW, were commonly linked with days to heading and maturity (Table 3 and Appendix A).

The associated markers were linked to various important annotated gene families. The detailed annotation and their location in the whole genome sequence of wheat are presented in Table 4. The genome-wide functional annotation revealed that the gene functions such as plant chitinases, NB-ARC and NBS-LRR, are associated with many annotated SNP markers. A few other gene annotations—peroxidase superfamily and Cytochrome P450, appear to show a positive role in NAD(P) H-based regulation of oxidoreductase activity during the hypersensitive response (Table 3).

## 3. Discussion

Biotic stresses such as spot blotch and abiotic, which are mainly terminal heat, challenge field realities while cultivating wheat in South Asia. Spot blotch and heat stress at post-anthesis become critical during grain filling; hence this stage needs special protection [8,19]. The high temperature during grain filling stages affects photosynthesis and slashes the yield [20,21]. Recently, some wheat genotypes have been identified as being tolerant to abiotic and biotic stresses [2,9,22], and new varieties are being released to sustain wheat production. This has been mainly achieved through screening materials under heat stress and disease nurseries, which is costly and time-consuming. The multi-location shuttle breeding strategy has proven helpful and successfully selected the most favourable alleles contributing to resistance/tolerance toward important stresses [2,23]. The kernel size and grain yield are affected by heat stress and spot blotch events near anthesis [1,2]. The simulated reduction in kernel size of up to 3% per degree Celsius rise in temperature is well within the range of 2–7% from field experiments [24]. Likely, the loss in the green area due to spot blotch and terminal heat affects grain size due to the less remobilization of water-soluble carbohydrates stored in stem and leaf sheaths to developing grains under high temperature and disease [25].

We identified a group of seven SNP markers associated with six phenotypic traits that control the combined and individual stress of spot blotch and terminal heat. Additionally, several QTLs were identified for the grain attributes, such as higher TGW, grain weight/per spike, spikelet number/per spike, grain size and grain area. NDVI, which was first used to map spot blotch resistance by Kumar et al. [16], who mapped the resistance locus Sb2 and reported a negative correlation between the NDVI and AUDPC, which was also confirmed in the present study. Markers associated with NDVI can be effectively used to select resistant genotypes with most of the fitness traits. NDVI is influenced by the days to heading and days to maturity. Therefore, a marker–trait association for days to heading and maturity, TKW, and yield depend on healthy leaf area measured as NDVI. Markers associated with these traits can be essential in selecting promising spot blotch-resistant genotypes with higher yields under heat-stressed environments. In synthetic hexaploids derived from *Ae. tauschii*, Okamoto et al. [26] identified QTLs responsible for grain size and shape variation in the D genome.

Similarly, Williams and Sorrells [27] (2014) reported 31 QTLs for Seed size and shape in Synthetic W7984 × Opata M85 (SynOpDH) population. Additionally, environmentally stable QTLs on 1A and 2D and a pleiotropic QTL on 5A were also detected. Recently, Yan et al. [13] extensively studied the genetic factors in the 2D and 7D controlling grain size and shape variation. Similarly, Kumari et al. [14] identified seven markers associated with grain area, days to heading, days to maturity, SPAD, and test weight (TGW), indicating the important genomic regions associated with these traits.

Gene annotation of 21 SNP markers linked to the spot blotch and terminal heat-associated traits was also identified. The SNP 3026360 on chromosome 2D was associated with NBS-LRR and S/TPK protein; these are the most common R-gene. Another maker, 1125940 on chromosome 1A, was annotated to the potato virus X resistance protein (RX), and Peptidase S8, subtilisin, Asp-active site that took part in the resistance against potato virus X and belongs to an N-terminal coiled-coil domain, a nucleotide-binding domain, and leucine-rich repeats (CC-NB-LRR) [28,29]. One more SNP marker, 1079395 (chromosome 1A), was annotated to peroxidase superfamily protein. This protein plays a role in self-defence [30] by catalyzing oxide reduction of H_2_O_2_. Moreover, it has multiple tissue-specific functions during the hypersensitive response (HR).

The SNP 1122111 on chromosome 5A is annotated to plant phospholipase D (PLD), a calcium-dependent enzyme. This enzyme is linked with drought tolerance [31]. Similarly, another SNP marker, 1395486, on chromosome 2A, was annotated to cysteine peptidases belonging to the papain-like cysteine peptidase. This superfamily involved programmed cell death (PCD) based on disease resistance in various pathosystems [32]. Few markers were associated with the EF-hand motif, calcium-binding domains, and Cytochrome P450, which has a positive role in NAD (P) H based on the regulation of oxidoreductase activity during the hypersensitive response. A study by Ayana et al. [33] identified genomic regions on chromosomes 2D, 5A, and 7B linked to NBS-LRR, S/TPK, and many plants’ defence-related protein families as Chitinase class I and peroxidases for spot blotch resistance.

Another gene with Zinc finger CCCH domain-containing protein pathogen-associated molecular pattern (PAMP)–that triggers immune responses was found in the genomic regions of SNP 1029559|F|0 and 1029767|F|0 in *Arabidopsis thaliana* [34].

In the genomic area of the SNP 995480|F|0 and wheat, two genes coding for Cytochrome P450 were also identified. The cysteine protease coding gene in the area is especially crucial since extracellular cysteine protease is required for pathogen recognition. Stress recognition causes an oxidative burst, followed by transcriptional reprogramming and HR, resulting in disease resistance [35]. Six F-box family proteins were also found in the region (SNPs 1034888|F|0, 1079395|F|0, 1045022|F|0, 1088945|F|0, and 3028841|F|0). F-box family protein controls various biological processes, including leaf senescence and responses to biotic [36] and abiotic stresses [37] independent of SAR via the ubiquitin–proteasome pathway. A ubiquitin family protein gene was discovered spanning the SNPs 2275693|F|0 and 1029767|F|0. Ubiquitin and associated proteins, which are components of the ubiquitin–proteasome system (UPS), regulate a variety of pathways, including responses to biotic and abiotic stimuli [38], and are one of the most important systems in plant defence [39].

In the backcross introgression lines produced from *T. durum* (cv. PDW274 susceptible) and *Ae. speltoides*, Kaur et al. [40] discovered five QTLs connected to SB resistance: Q.Sb.pau-2A, Q.Sb.pau-2B, Q.Sb.pau-3B, Q.Sb.pau-5B, and Q.Sb.pau-6A. The functional annotations for the previously published genomic regions are identical to those in the current work. At the same time, Tomar et al. [41] identified four new QTLs on Chr. 1A, 1D, 2B, and 6D that are associated with NBS-LRR, MADS-box transcription factors, and other disease-resistance protein families. Additionally, stable QTLs were detected on chromosomes 1B, 5A, 5B, 6A, 7A, and 7B in the CC population, explaining 2.89–10.32% of PV and collectively 39.91% of the total PV [42,43]. The quantitative genetic control of the spot blotch resistance, including markers linked to the *Lr46*, *Sb1*, *Sb2* and *Sb3* genes, has been reported recently [44]. The association of the 2NS translocation from *Ae. ventricosa* with spot blotch resistance and the spot blotch favourable alleles at the 2NS translocation, along with two markers on chromosome 3BS (3B_2280114 and 3B_5601689), has been reported first time from the multiple environment studies from Mexico and India. The findings of this study indicate the possibility of using the SNP linked for multiple stress regimes.

## 4. Materials and Methods

### 4.1. Plant Material, Experimental Design, and Layout of the Experiment

The experiments were conducted for three years (2014–2017) during the main wheat growing season (*Rabi/winter* season) at the Agricultural Research Farm of Banaras Hindu University, Varanasi (25.2° N and 83.0° E). One hundred eighty-five recombinant inbred lines (F_10_) of ‘*T. aestivum* (HUW 234) × *T. spelta* (H^+^26)’ cross and their parents were evaluated for spot blotch, terminal heat stress, and their combined effect under field conditions. This is the same population that Pandey et al. [1] used from the same institution—Banaras Hindu University.

The experiment was conducted using an incomplete lattice design with four replications under two different environments—the third week of November was considered as timely sown (no terminal heat stress) but favourable for spot blotch only (EN1). The next sowing was carried out in the last week of December, considered late sown and favourable for both—spot blotch and terminal heat (EN2) [21,45]. The experiment was plated in plots of 1.2 m × 2 rows at a 22 cm distance between the rows. The plot area was considered to be 0.5 m^2^. Approximately 50 seeds per row were sown. The detailed layout of the experiment is presented in Table 1. The crop was grown following prescribed agronomic practices (120 kg N: 60 kg P_2_O_5_: 40 kg K_2_O per hectare) along with four irrigations. Two replications in each year/environment were protected with fungicide (Azoxystrobin 125 a.i. g/h), while two replications were inoculated with an aggressive isolate of *B. sorokiniana*. Fungicide was applied twice in GS 45 and GS 65 on Zadok’s scale [46].

### 4.2. Pathogen Isolate and Inoculations

The *B. sorokiniana* isolate HD 3069 (MCC-1572) was multiplied by culturing on sorghum grain, following Chand et al. [47]. The spore suspensions were 10^4^/mL in water containing 0.1 mL/L Tween 20. Plants were sprayed in the evening at growth stage ZGS 55 [46], and the field was irrigated the same day for optimal disease development.

### 4.3. Phenotyping for the Assessment of Spot Blotch and Terminal Heat Stress

#### 4.3.1. Assessment of Disease Components

Scoring for disease reaction was initiated as soon as the first symptoms had appeared on all the accessions. The second scoring was conducted at ZGS 69, and the final was at ZGS 77. The scoring was conducted using a double-digit scale [48,49]. A disease severity (DS) index was calculated from the ratio (D1/9) × (D2/9) × 100. AUDPCs were derived from the DS, as outlined by Shaner and Finney [50,51], based on the expression
(1)$$AUDPC=\sum_{i=0}^{n=1}[\left\{\left(Y_{i}+Y\left(i+1\right)\right)÷2\right\}\times (t\left(i+1\right)-t_{i})]$$
where *y_i_* is an assessment of disease at the *i*th observation, *t_i_* is time (in days) at the *i*th observation, and *n* is the total number of observations.

#### 4.3.2. Estimation of Chlorophyll Content by Soil Plant Analysis Development (SPAD)

A Minolta SPAD-502 m (Minolta Camera Ltd., Osaka, Japan) was used for the non-destructive assessment of leaf chlorophyll content described by Schlemmer et al. [52]. SPAD value was obtained as the mean of three measurements (base, middle, and apex) of the flag leaf (F). Three plants were recorded for each line in each replication. SPAD values were recorded 14 days after inoculation (dai), and at 21 dai, and an average was determined.

#### 4.3.3. Canopy Temperature (CT)

The infrared gun LT 300 IRT was used to record CT; the readings were noted between 11:00 h to 14:00 h on cloudless, bright days within 0–4 days of disease assessment in the treated plots [53]. Canopy temperature was recorded at 14 dai and 21 dai and then averaged.

#### 4.3.4. Normalized Difference Vegetative Index (NDVI)

A hand-held GreenSeeker crop device (Trimble Navigation Ltd., Sunnyvale, CA, USA) was used to measure NDVI [10]; the readings were obtained between 11.00 and 14.00 h. within 0–4 days of disease assessment in the treated plots.

#### 4.3.5. Phenological Traits

Days to heading and physiological maturity (when the peduncle became yellow) were recorded from each RIL in each environment. The weight of 1000 kernels of individual RIL in each environment and each treatment was also recorded.

#### 4.3.6. Grain Scan for Measurement of Grain Area and Perimeter

A grain scan tool was used to measure the grain size and area [54]. For further analysis, the grain scan generated data on grain area (mm^2^) and perimeter (mm).

### 4.4. Genetic Analysis of Spot Blotch and Heat Stress Associated with Phenotypic Traits

#### 4.4.1. Genotyping

The genomic DNA was extracted from 21-day-old seedlings of 185 RILs, and their parents using the Diversity Array Technology protocol described online http://www.diversityarrays.com/sites/default/files/pub/DArT_DNA_isolation.pdf (accessed on 10 November 2015). The resulting DNA was used for SNP and DArT array through Diversity Arrays Technology Pty. Ltd. University of Canberra, Australia. The 13,460 single nucleotide polymorphism (SNP) and 14,791 DArT loci obtained [55] were used for genome-wide association studies (GWAS) of various phenotypic traits associated with spot blotch and heat stress.

#### 4.4.2. Population Structure Analysis

Population structure (Q) was analyzed using a model-based clustering method named STRUCTURE [56]. The number of subgroups (ΔK) in the panel was estimated following [57]. The fixation index (F_ST_) of subpopulations was obtained through STRUCTURE run outputs. Population Matrix Q was also obtained for further analysis. Model-based cluster analysis implemented in STRUCTURE determines LnPD values for grouping 185 wheat genotypes into distinct groups. These values were used to determine the number of genetically distinct sub-populations implemented in the web-based tool Structure Harvester [58].

#### 4.4.3. Genome-Wide Marker–Trait Association Analysis

The TASSEL 5.0 program [59] was used to calculate the population Kinship matrix based on the scaled identity by state (IBS) method using marker data that had passed quality filtering. Significant marker–trait associations (MTAs) were identified using a Mixed Linear Model (MLM) in TASSEL 5.0 (http://www.maizegenetics.net/; accessed on 20 May 2022) [59]. The analysis was carried out in PLINK [60], TASSEL [59], DARWIN [61,62] and GAPIT platforms in sequential order. The analysis was performed with a compressed mixed linear model [63] implemented in the GAPIT R package [64]. The MLM was run with the optimum compression level and previously determined population parameters [65]. To overcome the limitations of linkage mapping, LD mapping, a complementary strategy based on the correlation of genotype with phenotype in domesticated and natural populations, was used. This aided in shifting the emphasis from families to populations. The underlying principle of this approach is that LD between linked loci must be maintained over many generations. Linkage disequilibrium mapping exploits all historical recombination events in the population since the origin of the marker–trait association. However, to reduce the possibility of false positives in LD mapping. The population structure (Q) was estimated and then used in a mixed linear model to test for associations. The kinship relationships of the samples were also estimated for better control of type I error rates in association mapping, which accounts for population structure and relatedness.

#### 4.4.4. In-Silico Analysis

The physical starting point of the marker preceded by the chromosome name was brought to Ensembl. A few thousand base pairs were added before and after (e.g., if the marker’s position was 943389 on chromosome 2A, we used 2A: 942423–946423) to find the candidate genes linked to significant markers. The number of base pairs added varied for each marker depending on its proximity to the genes, but only the genes in the same genetic position were considered. The interval was then explored for predicted genes, and annotations from the IWGSC (https://www.wheatgenome.org/ accessed on 5 June 2022) were obtained. For several genes, the IWGSC annotations were not available. So, they evaluated based on orthologous genes in related species with known predicted functions using the comparative genomics tool in Plant Ensembl. In some cases, when the genes had a less similar disease resistance orthologue (<70%) in the annotated genomes of the related species in Ensembl, the sequence of the *T. aestivum* gene was brough to NCBI. The nucleotide basic local alignment search tool (BLAST) (http://blast.ncbi.nlm.nih.gov/Blast.cgi accessed on 5 June 2022) was used where only highly similar sequences (mega-blast) were considered. This search also included the gene predictions in different species available in GenBank but not in Ensembl. The *T. aestivum* gene transcripts and their available domains in Ensembl were also used (using the show transcript table link).

The blast (https://wheat.triticeaetoolbox.org/tools/blast/ accessed on 5 June 2022) in the Triticeae Toolbox website was used to perform a nucleotide BLAST (BLAST-n) of the significant marker sequences against the GBS markers in the Triticeae Toolbox (T3) database. Moreover, the JBrowse tool from T3 and GBrowse from URGI (https://urgi.versailles.inra.fr/gb2/gbrowse/wheat_survey_sequence_annotation; accessed on 25 May 2022) was also used to identify annotation to SNP markers.

### 4.5. Statistical Analysis

The statistical analysis was carried out using SAS software (version 9.2) [64]. The Sapiro and Wilnks test was first used to assess the normality of data, and the homogeneity of variance was determined using the Levene test. Field data from three consecutive years were subjected to variance analysis to determine significant differences among treatments using PROC GLM and the mixed model of SAS software. Correlation among the variables was established by PROC CORR using replicated data, and Bonferroni’s adjustments at *p* = 0.05 were used to differentiate and group the genotype based on different variables.

## 5. Conclusions

Spot blotch and terminal heat tolerance are major constraints on wheat harvest, particularly in hot and humid climates prevailing in South Asia. Terminal heat and spot blotch lead to premature leaf senescence, reduced grain filling, low kernel weight, and reduced yield. The new sources of resistance must be continually identified and introgressed to counteract the restrictions posed by these stresses. The current work sheds light on the genetic regions that confer resistance to the combined stress of spot blotch and terminal heat stress. This research also specifies the possible use of NDVI, canopy temperature, and gain characteristics as indicator characteristics for high-throughput screening for these stresses during the vegetative and grain-filling stages. The genomic domains annotated to Zinc finger domains, cysteine protease coding gene, F-box family protein, ubiquitin and related proteins, and Cytochrome P450 reveal a significant role in the combined stress of spot blotch and terminal heat in bread wheat. The study also emphasizes *T. speltoides* as a source of resistance to spot blotch and terminal heat tolerance.

## Figures and Tables

**Figure 1 plants-11-02987-f001:**
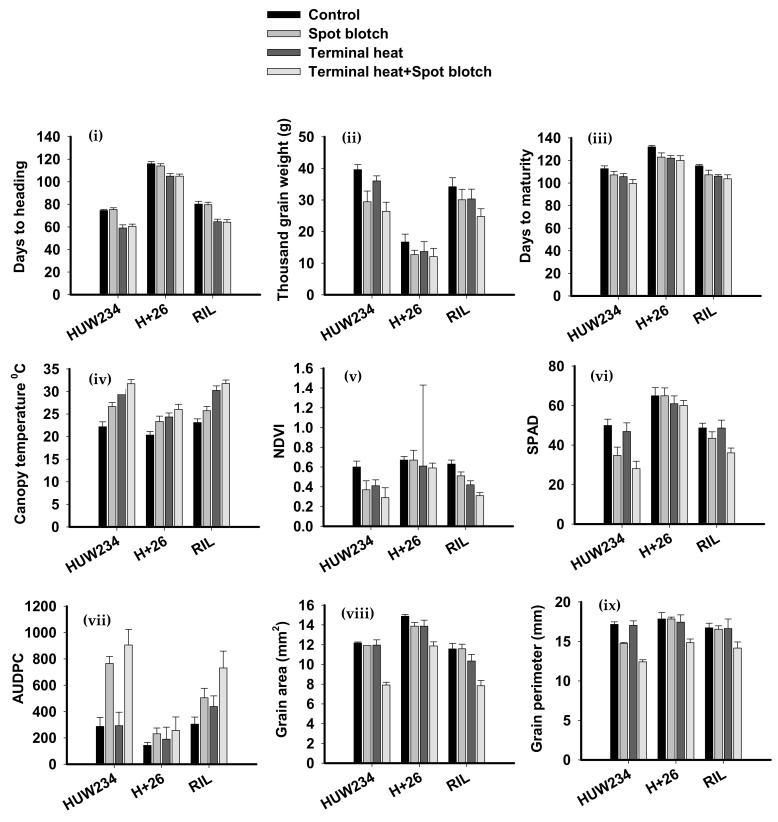
Summary of the effect of biotic and abiotic stresses on nine quantitative traits: (**i**) days to heading, (**ii**) thousand kernel weight, (**iii**) days to maturity, (**iv**) canopy temperature, (**v**) normalized distributed vegetation index (NDVI), (**vi**) soil plant analysis development (SPAD), (**vii**) area under disease progress curve (AUDPC), (**viii**) grain area, and (**ix**) grain perimeter.

**Figure 2 plants-11-02987-f002:**
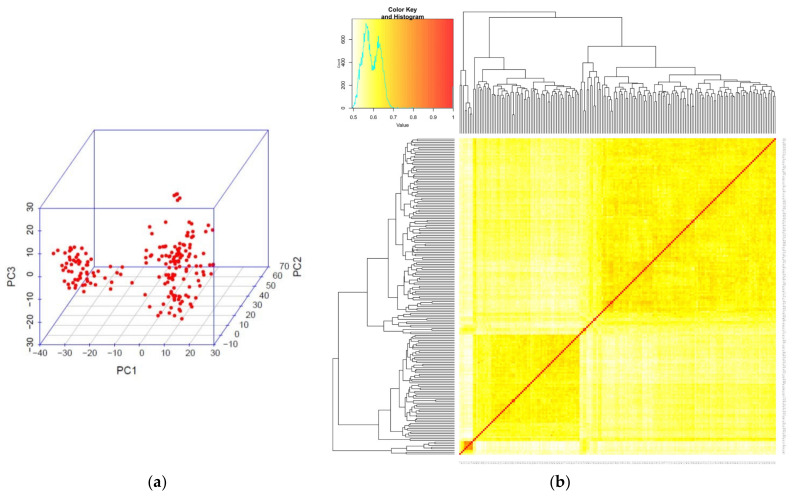
(**a**) Three-dimensional plot of the first three principal components showing the genetic differences among 185 RILs and parents. (**b**) The heat map developed from 6734 SNP markers showed clustering of 185 RILs and parents.

**Figure 3 plants-11-02987-f003:**
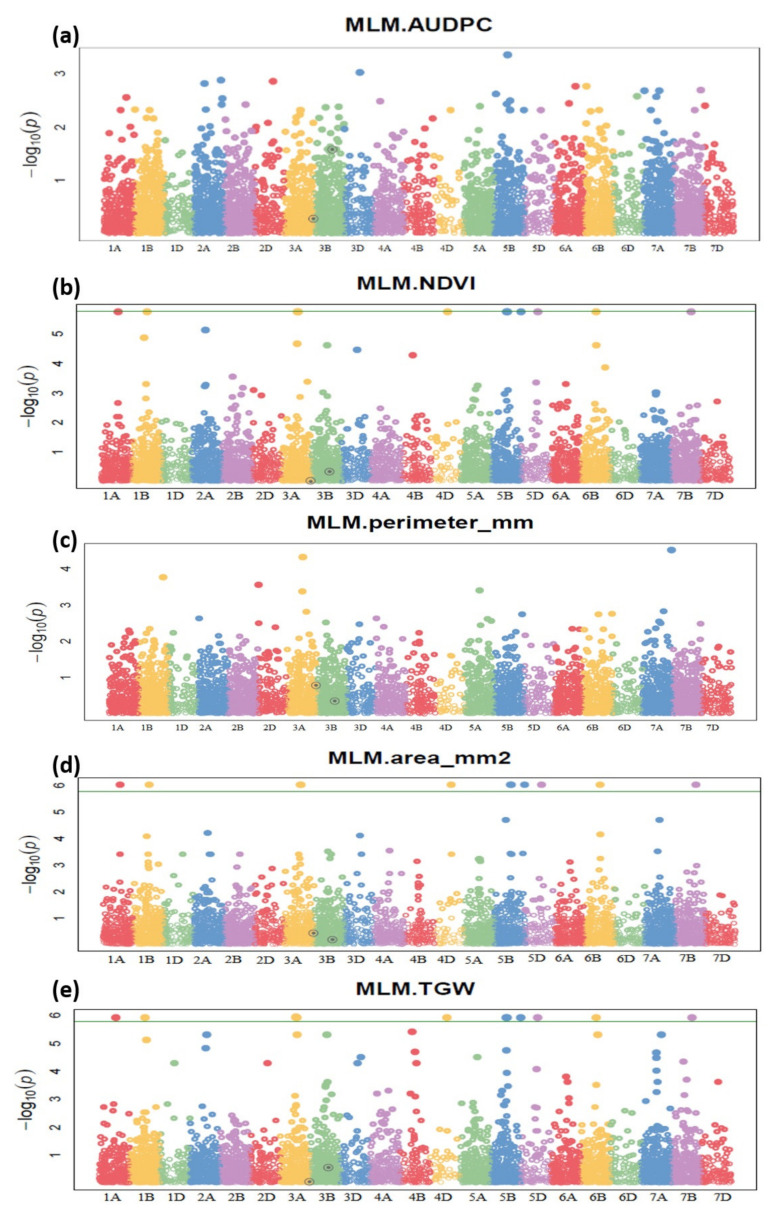
Genome-wide association scan for (**a**) area under disease progress (AUDPC), (**b**) normalized distributed vegetation index (NDVI), (**c**) grain perimeter, (**d**) grain area, (**e**) thousand-grain weight in RILs. The Manhattan plot was developed using a mixed linear model (MLM). The −log10 (*p*) values from a genome-wide scan are plotted against positions on each of the 21 wheat chromosomes. Horizontal lines indicate genome-wide significance thresholds.

**Table 1 plants-11-02987-t001:** Effect of biotic and abiotic stress on the performance of nine traits in wheat.

Sr. No.	Phenotypic Trait	Control	Spot Blotch	Terminal Heat Stress	Spot Blotch + Terminal Heat Stress
1	Days to heading (days)	90.23 ± 1.79	89.71 ± 1.92	76.2 ± 2.46	76.57 ± 2.01
2	Thousand-grain weight (g)	30.15 ± 2.32	24.06 ± 2.71	26.68 ± 2.6	21.07 ± 2.65
3	Days to Maturity (days)	119.95 ± 1.66	112.49 ± 3.65	111.26 ± 2.06	107.75 ± 3.82
4	Canopy Temperature (°C)	21.88 ± 0.9	25.24 ± 1.01	28.36 ± 0.91	29.84 ± 0.94
5	NDVI	0.63 ± 0.05	0.52 ± 0.08	0.48 ± 0.09	0.4 ± 0.06
6	SPAD	54.5 ± 3.24	47.7 ± 3.82	52.14 ± 4.11	41.37 ± 2.91
7	AUDPC	244.69 ± 48.09	500.28 ± 56.39	306.91 ± 91.36	630.62 ± 116.58
8	Grain Area (mm^2^)	12.87 ± 0.28	12.45 ± 0.42	12.05 ± 0.6	9.21 ± 0.41
9	Grain Perimeter (mm)	17.24 ± 0.57	16.37 ± 0.26	17.03 ± 0.89	13.8 ± 0.5

# Data is mean ± SD.

**Table 2 plants-11-02987-t002:** Analysis of variance for nine traits during the interaction of various treatments and environments (2015–2018).

Source	DF	Mean Sum of Squares								
		DH	TKW	DM	CT	NDVI	SPAD	AUDPC	Grain Area	Grain Perimeter
Year	2	36.71 *	9301.43 *	1515.34 *	5085.82 *	2.93 *	1275.91 *	19,890,523.95 *	38,070.98 *	32,605.08 *
Condition	1	266,584.51 *	23,293.19 *	44,301.94 *	47,917.63 *	47.16 *	15,318.28 *	35,978,424.99 *	6833.56 *	1701.23 *
Treatment	1	242.67 *	25,999.75 *	29,070.34 *	4757.77 *	15.16 *	88,172.20 *	67,495,554.29 *	1716.54 *	1928.05 *
RILs	184	81.23 *	116.31 *	52.74 *	6.082 *	0.0094 *	67.52 *	80,137.94 *	3.33 *	5.78 *
Rep	1	26.34	802.07 *	19.6	118.02 *	0.00011	962.57 *	1,885,835.47 *	1.29	40.88 *
Year × RILs	368	1.55	21.63 *	8.31 *	2.13 *	0.00345 *	8.86	44,840.46 *	0.632 *	0.52
Condition × RILs	184	8.58 *	18.05 *	44.91 *	4.26 *	0.0051 *	49.02 *	24,927.34 *	1.01 *	3.12 *
Treatment × RILs	184	36.72 *	56.38 *	48.92 *	4.92 *	0.00706 *	56.99 *	57,215.63 *	2.08 *	4.10 *
Year × Condition × Treatment × RILs	1295	2.04	19.51 *	21.52 *	6.50 *	0.0111 *	80.076 *	48,598.18 *	5.20 *	8.93 *

* significant at *p* < 0.0001. DF: degrees of freedom, DH: days to heading, TKW: thousand kernel weight DM: days to maturity, CT: canopy temperature, NDVI: normalized distributed vegetative index, AUDPC: area under disease progress curve, SPAD: soil plant analysis development.

**Table 3 plants-11-02987-t003:** Significant SNP and annotated proteins and transcripts on the high confidence genes based on wheat reference genome RefSeq1.1 (Ensembl Plant release 50, IWGSC RefSeq v1.1, October 2022).

Sr	Markers	Genomic Location	Trait	Transcript ID	Description
1	1058939|F|0	1A:530168043–530267926	NDVI	TraesCS1A02G340000	IPR027145: Periodic tryptophan protein 2, IPR020472: G-protein beta WD-40 repeat
				TraesCS1A02G339900	IPR003960: ATPase, AAA-type, conserved site, CDC48, IPR003338: N-terminal subdomain
				TraesCS1A02G339800	IPR006852: Glycosyltransferases, MUCI70
				TraesCS1A02G340100	IPR027417: P-loop containing nucleoside triphosphate hydrolase, IPR018368: ClpA/B, conserved site 1
2	1102573|F|0	1A:403359429–403388874	NDVI	TraesCS1A02G230600	Kinesin-like protein KIN-7N, IPR027417: P-loop containing nucleoside triphosphate hydrolase
3	1125940|F|0	1A: 499801953–499801988	Grain area, Days to heading, days to maturity, SPAD, Test weight (TGW), Canopy temperature	TraesCS1A02G309000	A0A1D5RW30 IPR001611: Leucine-rich repeat IPR013210: Leucine-rich repeat-containing N-terminal, plant-type IPR032675: Leucine-rich repeat domain superfamily
				TraesCS1A02G308800	A0A341NPS0 IPR002048: EF-hand domain IPR011992: EF-hand domain pair IPR018247: EF-Hand 1, calcium-binding site IPR039030: Calmodulin
				TraesCS1A02G308900	IPR032640: AMP-activated protein kinase, glycogen-binding domain
4	1241625|F|0	1B:687710001–687804688	Grain perimeter	TraesCS1B02G480300	IPR013210: Leucine-rich repeat-containing N-terminal, plant-type, IPR017441: Protein kinase, ATP binding site
				TraesCS1B02G480100	IPR012337: Ribonuclease H-like superfamily, IPR014811: Argonaute, linker 1 domain
				TraesCS1B02G480400	IPR045877: RNA-binding protein ZFP36-like, IPR000571: Zinc finger, CCCH-type
				TraesCS1B02G480200	IPR003388: Reticulon-like protein B12
5	1395486|F|0	1B:315459354–315462324	Grain Area, Days to heading, days to maturity, SPAD, Test weight (TGW), Canopy temperature	TraesCS1B02G174800	A0A341PBP3 IPR001210: Ribosomal protein S17e IPR036401: Ribosomal protein S17e-like superfamily
				TraesCS1B02G174900	A0A1D5SGD8 IPR001107: Band 7 domain IPR027705: Flotillin family IPR036013: Band 7/SPFH domain superfamily
6	3064765|F|0	1B:683455649–683585555	NDVI	TraesCS1B02G474800	IPR044798, Chromatin modification-related protein EAF1A/B
				TraesCS1B02G475200	IPR032675: Leucine-rich repeat domain
				TraesCS1B02G475400	IPR004907, ATPase, V1 complex, subunit C
7	1077356|F|0	2A:617631545–617741244	Days to heading, Days to maturity	TraesCS2A02G375100	IPR004182: GRAM domain, GEM-like protein
				TraesCS2A02G374800	IPR030847: Mitochondrial glycine transporter Hem25/SLC25A38
8	2281188|F|0	2A:42147190–42231485	AUDPC	TraesCS2A02G088800	IPR000719: Protein kinase domain IPR001245: Serine-threonine/tyrosine-protein kinase, catalytic domain
				TraesCS2A02G088900	IPR042449: Ubiquitin-activating enzyme E1, inactive adenylation domain, subdomain 1 THIF-type NAD/FAD-binding (IPR000594)
9	1039495|F|0	2A:703378875–703411784	AUDPC	TraesCS2A02G454500	IPR013088, Zinc finger, NHR/GATA-type
				TraesCS2A02G454600	IPR044533, FCS-Like Zinc finger 1/2/3
10	2253029|F|0	2A:718816359–718988414	NDVI	TraesCS2A02G482000	CASP-like protein, IPR006702: Casparian strip membrane protein
				TraesCS2A02G482100	IPR001509, NAD-dependent epimerase/dehydratase
				TraesCS2A02G482400	IPR044661, Mediator of RNA polymerase II transcription subunit 15a/b/c-like
				TraesCS2A02G482700	IPR029063: S-adenosyl-L-methionine-dependent methyltransferase
11	3028841|F|0	2D:648337989–648380136	AUDPC	TraesCS2D02G594900	F-box domain-containing protein-related
				TraesCS2D02G594800	IPR000504: RNA recognition motif, IPR012677: Nucleotide-binding alpha-beta plait domain
				TraesCS2D02G594700	RING-type E3 ubiquitin transferase, IPR013083: Zinc finger
12	1019339|F|0	2D:625109256–625163527	Grain perimeter	TraesCS2D02G549600	IPR027417: P-loop containing nucleoside triphosphate hydrolase, IPR042197: Apoptotic protease-activating factors, helical domain
				TraesCS2D02G549700	IPR032675: Leucine-rich repeat domain superfamily, IPR044974: Disease resistance protein, plants
13	983670|F|0	3A:640746667–640773225	Days to heading, Days to maturity	TraesCS3A02G392900	IPR000109: Proton-dependent oligopeptide transporter, IPR018456: PTR2 family proton/oligopeptide symporter, conserved site
				TraesCS3A02G393000	IPR029058: Alpha/Beta hydrolase fold, IPR002168: Lipase, GDXG, putative histidine active site
14	3064641|F|0	3A:13351697–13418670	Grain perimeter	TraesCS3A02G025000	IPR023213: Chloramphenicol acetyltransferase-like domain
				TraesCS3A02G025300	Zinc finger FYVE domain-containing protein, IPR035669: GDSL lipase/esterase-like, plant
				TraesCS3A02G025200	E3 ubiquitin-protein ligase, IPR013083: Zinc finger, RING/FYVE/PHD-type
15	2256281|F|0	3A: 512307520–512555787	Grain area, Days to heading, days to maturity, SPAD, Test weight (TGW), Canopy temperature	TraesCS3A02G284100	Hexosyltransferase
				TraesCS3A02G283600	Peptidase S8 propeptide/proteinase inhibitor
				TraesCS3A02G283700	IPR007608: Senescence regulator S40
				TraesCS3A02G283900	IPR013857: NADH: ubiquinone oxidoreductase intermediate-associated protein 30
				TraesCS3A02G284100	Hexosyltransferase IPR002495: Glycosyl transferase, family 8 IPR029044: Nucleotide-diphospho-sugar transferases IPR029993: Plant galacturonosyltransferase GAUT
				TraesCS3A02G284200	Protodermal factor 1
16	980238|F|0	3A:638969536–639257641	Grain Area, Days to heading, days to maturity, SPAD, Test weight (TGW), Canopy temperature	TraesCS3A02G390800	tetratrico peptide repeat region (TPR)
				TraesCS3A02G390900	IPR007234:Vps53-like, N-terminal IPR039766: Vacuolar protein sorting-associated protein 53
				TraesCS3A02G391000	IPR025993: Ceramide glucosyltransferase IPR029044: Nucleotide-diphospho-sugar transferases
				TraesCS3A02G391100	IPR029768: Fructose-bisphosphate aldolase class-I active site
				TraesCS3A02G391400	IPR017907: Zinc finger, RING-type, conserved site
17	2275693|F|0	3A:647604415–647821505	Days to heading, Days to maturity, Test weight	TraesCS3A02G402100	IPR011009: Protein kinase-like domain, IPR036426: Bulb-type lectin domain
				TraesCS3A02G402200	IPR001763: Rhodanese-like domain
				TraesCS3A02G402300	AS2, IPR004883: Lateral organ boundaries, LOB
18	976829|F|0	3B:672380580–672583235	Days to heading, Days to maturity	TraesCS3B02G433400	A0A077S6B7: BTB/POZ and TAZ domain-containing protein 3
				TraesCS3B02G433500	IPR018247: EF-Hand 1, calcium-binding site
				TraesCS3B02G433600	IPR001245: Serine-threonine/tyrosine-protein kinase, catalytic domain, IPR000719: Protein kinase domain, IPR008271: Serine/threonine-protein kinase, active site
				TraesCS3B02G433900	IPR013210: Leucine-rich repeat-containing N-terminal, plant-type
19	1088945|F|0	3D:436639761–436684217	AUDPC	TraesCS3D02G323600	IPR032675: Leucine-rich repeat domain, IPR044997 F-box protein, plant
				TraesCS3D02G323700	DNA-(apurinic or apyrimidinic site) lyase, IPR005135: Endonuclease/exonuclease/phosphatase
				TraesCS3D02G323800	FK506-binding-like protein (PTHR34567)
20	1034888|F|0	4A:597665575–598072618	Canopy temperature	TraesCS4A02G298500	IPR036410: Heat shock protein DnaJ, cysteine-rich domain
				TraesCS4A02G298600	IPR002068: Alpha crystallin/Hsp20 domain IPR008978:HSP20-like chaperone
				TraesCS4A02G299400	IPR002347: Short-chain dehydrogenase/reductase, IPR036291: NAD(P)-binding domain
				TraesCS4A02G298700	IPR013215: Cobalamin-independent methionine synthase MetE,
21	1050819|F|0	4D:46589710–46738670	Grain area, Days to heading, days to maturity, SPAD, Test weight	TraesCS4D02G071900	IPR044837 B3 domain-containing protein REM16-like, IPR015300 DNA-binding pseudo barrel domain superfamily
				TraesCS4D02G072000	IPR001461: Aspartic peptidase A1, IPR034161: Pepsin-like domain, plant
22	1088359|F|0	5A:503866360–503955851	Grain perimeter	TraesCS5A02G295400	IPR017736: Glycoside hydrolase, family 1, beta-glucosidase
				TraesCS5A02G295800	IPR003527: Mitogen-activated protein (MAP) kinase, conserved site IPR008271: Serine/threonine-protein kinase, active site
23	1029767|F|0	5A:615146872–615353569	NDVI	TraesCS5A02G431100	IPR007275: YTH domain
				TraesCS5A02G431600	Casein kinase I, photoperiodic control of flowering time, long-day repression, IPR008271: Serine/threonine-protein kinase, active site
				TraesCS5A02G431500	IPR032675: Leucine-rich repeat domain superfamily
				TraesCS5A02G431300	IPR001810: F-box domain IPR032675: Leucine-rich repeat domain
24	1045022|F|0	5A:691658614–691905149	NDVI	TraesCS5A02G534500	Flavin-containing monooxygenase, IPR036188: FAD/NAD(P)-binding domain superfamily
				TraesCS5A02G534800	IPR001810: F-box domain
				TraesCS5A02G534900	IPR042101: Signal recognition particle SRP54, IPR027417:P-loop containing nucleoside triphosphate hydrolase
				TraesCS5A02G534200	IPR039605: AT-hook motif nuclear-localized protein
25	3064380|F|0	5A:27509863–27509903	NDVI	TraesCS5A02G042600LC	NA
				TraesCS5A02G042700LC	NA
26	1126383|F|0	5B:568398994–568517930	Days to heading, Days to maturity, SPAD, Test weight, Grain area	TraesCS5B02G389200	IPR002885: Pentatricopeptide repeat IPR011990: Tetratricopeptide-like helical domain
				TraesCS5B02G389300	EDA15, R022192: Mitochondrial degradosome RNA helicase subunit, C-terminal domain
				TraesCS5B02G389400	IPR044593, FCS-Like Zinc finger 8/MARD1
27	3064429|F|0	5B:596900954–596988713	AUDPC	TraesCS5B02G421900	IPR044991, Tetraspani, plant, auxin-activated signalling pathway
				TraesCS5B02G421100	IPR044659, Protein PELPK-like, Proline-rich protein 10, At5g09530
28	1029559|F|0	5B:281567207–281859354	Grain area, Days to heading, days to maturity, SPAD, Test weight (TGW)	TraesCS5B02G152400	IPR018247: EF-Hand 1, calcium-binding site IPR039647: EF-hand domain pair protein CML-like
				TraesCS5B02G152100	IPR029962 Trichome birefringence-like family
				TraesCS5B02G152300	IPR000547: Clathrin, heavy chain/VPS, 7-fold repeat IPR011990: Tetratricopeptide-like helical domain
				TraesCS5B02G152200	IPR014014: RNA helicase, DEAD-box type, Q motif IPR027417:P-loop containing nucleoside triphosphate hydrolase
				TraesCS5B02G152500	Ribosome assembly factor mrt4 IPR040637: 60S ribosomal protein L10P, insertion domain
				TraesCS5B02G152600	IPR017932 Glutamine amidotransferase type 2 domain
				TraesCS2D02G534800	IPR008271: Serine/threonine-protein kinase, active site
29	1020582|F|0	5B:609824667–609977091	Grain area, Days to heading, Days to maturity, SPAD, Test weight (TGW)	TraesCS5B02G435300	IPR002213: UDP-glucuronosyl/UDP-glucosyltransferase
				TraesCS5B02G435600	IPR043325: Alpha-Amylase Inhibitors (AAI), Lipid Transfer (LT) and Seed Storage (SS) Protein
30	987983|F|0	5D:104592141–104634865	Days to heading, Days to maturity, SPAD, Grain area	TraesCS5D02G095300	IPR001611: Leucine-rich repeat IPR008271: Serine/threonine-protein kinase, active site, IPR000719: Protein kinase domain
				TraesCS5D02G095400	IPR002171: Ribosomal protein L2 IPR008991: Translation protein SH3-like domain
31	2266275|F|0	6B:708055234–708286758	Days to heading, Days to maturity	TraesCS6B02G448700	IPR035896: AN1-like Zinc finger
				TraesCS6B02G447800	IPR044974, Disease resistance protein, plants IPR038005: Virus X resistance protein-like, coiled-coil domain
32	987210|F|0	6B:5683365–5845027	Days to heading, Days to maturity	TraesCS6B02G008700	IPR044814: Terpene cyclases, class 1, plant
				TraesCS6B02G008900	IPR008271: Serine/threonine-protein kinase, active site, IPR017441: Protein kinase IPR032675: Leucine-rich repeat domain
				TraesCS6B02G008800	IPR001232: S-phase kinase-associated protein 1-like
				TraesCS6B02G009105	IPR001881: EGF-like calcium-binding domain IPR008271: Serine/threonine-protein kinase, active site IPR011009: Protein kinase-like domain superfamily IPR018097: EGF-like calcium-binding, conserved site IPR025287: Wall-associated receptor kinase, galacturonan-binding domain
33	995480|F|0	6B:80769409–81048854	NDVI	TraesCS6B02G102800	IPR001810: F-box domain
				TraesCS6B02G102900	IPR008176: Defensin, plant, Amylase inhibitor-like protein
				TraesCS6B02G103200	IPR006813: Glycosyl transferase, family 17
34	1021511|F|0	7A:83081610–83137614	Days to heading, Days to maturity	TraesCS7A02G129000	IPR003311: AUX/IAA protein
35	2280866|F|0	7A:4249205–4264215	Grain perimeter	TraesCS7A02G009600	IPR023296, Glycosyl hydrolase, five-bladed beta-propellor domain
36	2278379|F|0	7B:134493827–134645674	Days to heading, Days to maturity	TraesCS7B02G115900	IPR032799: Xylanase inhibitor, IPR001461: Aspartic peptidase A1 family
				TraesCS7B02G116200	PTHR31989: NAC domain- containing protein 82
37	1079395|F|0	7B:666498423–666698769	NDVI	TraesCS7B02G399800	IPR022991: Ribosomal protein L30e, conserved site
				TraesCS7B02G400300	IPR023213: Chloramphenicol acetyltransferase-like domain

**Table 4 plants-11-02987-t004:** Experimental layout for three consecutive cropping seasons (2015 to 2018).

Year	Environment	Treatment	Population
2015–2018 Cropping seasons November to April	EN1: Timely sown (last week of November)	Treatment 1: Control (no pathogen inoculation, protected using fungicide)	185 RILs + Parents (2 replications)
		Treatment 2: Spot blotch (inoculation by spot blotch pathogen, no protection by fungicide)	185 RILs + Parents (2 replications)
	EN2: Late sown (last week of December)	Treatment 3: Terminal Heat stress (no pathogen inoculated/protected using fungicide)	185 RILs + Parents (2 replications)
		Treatment 4: Spot blotch + terminal heat stress (inoculation by spot blotch pathogen no protection by the fungicide)	185 RILs + Parents (2 replications)

## Data Availability

Raw phenotypic and genotypic data is submitted at “Mendeley Data” and available with the link, Mendeley Data, V1, https://doi.org/10.17632/k3ms7wmcjy.1 (accessed on 27 October 2022).

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
