# Peer review of "New Genomic Regions Identified for Resistance to Spot Blotch and Terminal Heat Stress in an Interspecific Population of Triticum aestivum and T. spelta"

_plants, 2022, doi:10.3390/plants11212987_

Round 1

Reviewer 1 Report

The article presents the evaluation of the characteristics of resistance to spot bloch and heat stress of recombinant wheat varieties. In a significant way, they establish a variety of characteristics that refer to the state of the plants and, based on these results, confirm the resistance or tolerance of the recombinant wheat varieties.

I consider that the analysis is very complete and that it allows a clear assessment of the response of plants to biotic and abiotic stress.

On the other hand, they show a correlation between the characteristics of the plant, its location in the genome and the identification and description of the transcripts associated with these traits. On the other hand, they identify various QTLs associated with the qualities of the grain, which once again opens an option for agricultural improvement.

In this aspect, the work acquires greater relevance by opening multiple options that lead to the selection of a greater number of varieties resistant to these stresses and that favor greater agricultural productivity.

Author Response

Dear Reviewer, 

Thanks for endorsing our manuscript and giving your valuable time; we have gone through the manuscript to read and correct any mistakes left behind in methodology and results section. 

Thanks

Reviewer 2 Report

The study describes potential SNPs associated with terminal heat and spot blotch in a population published earlier.  Both individual and combined stress were applied to describe the impact on yield-related parameters. The information is useful for those working in this area. 

The availability of raw data may be ensured by the authors. The manuscript needs a thorough read by an expert in the language.

Author Response

Dear reviewer, 

Thank you for endorsing the MS and giving your valuable time to read the draft. Regarding your concern about data availability, we have edited our data availability statement. We have uploaded our data (phenotyping and genotyping data) at "Mendeley Data" and available with link, Mendeley Data, V1, doi: 10.17632/k3ms7wmcjy.1. 

We also correction grammatical mistakes as well through reading for language corrections. 

I hope this address your concern. 

Thanking you, 

Reviewer 3 Report

The manuscript revealed genomic regions associated with tolerance to combined stress to spot blotch and terminal heat by using one hundred eighty-five recombinant lines derived from the interspecific hybridization of 'Triticum aestivum × T. spelta' and have got some novel findings. The results can be used in wheat breeding to improve the tolerance to spot blotch and terminal heat. I suggest that the manuscript be revised for publication. Here are a few issues to focus on in revising the manuscript.

1.     Two traits were studied in the manuscript. Timely sown and late sown were used to evaluate the tolerance to terminal heat stress. I’m wondering how to establish a correlation between sown date and terminal heat tolerance.

2.     A recombinant inbred lines population derived from the interspecific hybridization of 'Triticum aestivum × T. spelta' was used in molecular mapping for the tolerance to spot blotch and terminal heat. The QTL in such population can be mapped by using inbreeding linkage analysis methods, but this manuscript used linkage disequilibrium analysis methods. I suggest the author explain the reasons in the manuscript.

3.     Are the field experiments planted in plots, single row or one plant? What is the area of the plot? How long is the row?

4.     Estimation of chlorophyll content, canopy temperature, normalized difference vegetative index, days to heading and physiological maturity, the weight of 1000 kernels, grain scan and disease components were evaluated in the population. But the most correlation coefficients between them are not high, which trait can be used for evaluating the tolerance to combined stress to spot blotch and terminal heat? Yield is the most important trait in wheat production including the tolerance to stress, what are the results of plot yield in field experiment?

Author Response

Dear Reviewer,

Thank you for the useful comments. Those helped us to improve the manuscript.

  1. Two traits were studied in the manuscript. Timely and late sown were used to evaluate the tolerance to terminal heat stress. I’m wondering how to establish a correlation between sown date and terminal heat tolerance.

Response: Two different environments - the third week of November was considered as timely sown (no terminal heat stress) but favourable for spot blotch only (EN1). The next sowing was done in the last week of December, considered late sown and favourable for both - spot blotch and terminal heat (EN2). These two environments are decided based on the previous studies (Paliwal et al. 2012 and Tiwari et al 2013).  Based on the previous studies and weather data of location Varanasi, when sowing is done in 3rd week of Nov. the grain filling during falls in 1st -4th week of Feb can serve as a control for the terminal heat. While sowing dates on the last week of Dec lead to the grain filling in 3rd week of March to 2nd week of April when the temperature remains higher, and ideal for the terminal heat treatments under field conditions.

  1. A recombinant inbred lines population derived from the interspecific hybridization of 'Triticum aestivum × T. spelta' was used in molecular mapping for the tolerance to spot blotch and terminal heat. The QTL in such a population can be mapped by using inbreeding linkage analysis methods, but this manuscript used linkage disequilibrium analysis methods. I suggest the author explain the reasons in the manuscript.

Response: To overcome the limitations of linkage mapping, LD mapping, a complementary strategy based on the correlation of genotype with phenotype in domesticated and natural populations, was used. This aided in shifting the emphasis from families to populations. The underlying principle of this approach is that LD between linked loci must be maintained over many generations. Linkage disequilibrium mapping exploits of all historical recombination events in the population since the origin of the marker-trait association. However, to reduce the possibility of false positives in LD mapping. The population structure (Q) was estimated and then used in a mixed linear model to test for associations. The kinship relationships of the samples were also estimated for better control of type I error rates in association mapping, which accounts for population structure and relatedness.

The above information is updated in the manuscript as per suggestions.

  1. Are the field experiments planted in plots, a single row or one plant? What is the area of the plot? How long is the row?

Response: The experiment was plated in plots of 1.2m × 2 rows at 22 cm distance between the rows. The plot area was considered to be 0.5 m2. Approximately 50 seeds per row. Information is updated in the manuscript.

  1. Estimating of chlorophyll content, canopy temperature, normalized difference vegetative index, days to heading and physiological maturity, the weight of 1000 kernels, grain scan and disease components were evaluated in the population. But the correlation coefficients between them are not high; which trait can be used for evaluating the tolerance to combined stress to spot blotch and terminal heat? Yield is the most important trait in wheat production including tolerance to stress; what are the results of plot yield in field experiment?

Response: The traits Normalized distributed vegetation index (NDVI), canopy temperature (CT) thousand kernel weight, and Soil Plant Analysis Development (SPAD) were more descriptive under stressed conditions and can be used for the evaluations.  The vegetation index (NDVI) was reduced from 0.63 ± 0.04 to 0.34 ± 0.03 indicating depletion by 46.03 %. The decrease in grain area was observed from 11.57 ± 056 mm2 to 7.84 ± 0.53 mm2  (32.23 %) which was implicated in the reduction of thousand kernel weight from 34 .17 ± 2.85 gm to 24.75 ± 2.46 gm (27.56 %).